# The fracture resistance of 3D-printed versus milled provisional crowns: An in vitro study

**Ahmed Othman** [ID]*, **Maximillian Sandmair, Vasilios Alevizakos** [ID]**, Constantin von See**

Research Center for Digital Technologies in Dentistry and CAD/CAM, Department of Dentistry, Faculty of Medicine and Dentistry, Danube Private University, Krems, Austria

* ahmed.othman@dp-uni.ac.at

**Data Availability Statement:** The data underlying the results presented in the study are available from Danube Private University, Steiner Landstrasse 124, 3500, Krems an der Donau, Austria. https://www.dp-uni.ac.at/de/home

## Abstract

### Background

CAD/CAM has considerably transformed the clinical practice of dentistry. In particular, advanced dental materials produced via digital technologies offer unquestionable benefits, such as ideal mechanical stability, outstanding aesthetics and reliable high precision. Additive manufacturing (AM) technology has promoted new innovations, especially in the field of biomedicine.

### Aims

The aim of this study is to analyze the fracture resistance of implant-supported 3D-printed temporary crowns relative to milled crowns by compression testing.

### Methods

The study sample included 32 specimens of temporary crowns, which were divided into 16 specimens per group. Each group consisted of eight maxillary central incisor crowns (tooth 11) and eight maxillary molar crowns (tooth 16). The first group (16 specimens) was 3D printed by a mask printer (Varseo, BEGO, Bremen, Germany) with a temporary material (VarseoSmile Temp A3, BEGO, Bremen, Germany). The second group was milled with a millable temporary material (VitaCAD Temp mono-color, Vita, Bad Säckingen, Germany). The two groups were compression tested until failure to estimate their fracture resistance. The loading forces and travel distance until failure were measured. The statistical analysis was performed using SPSS Version 24.0. We performed multiple t tests and considered a significance level of p <0.05.

### Results

The mean fracture force of the printed molars was 1189.50 N (±250.85) with a deformation of 1.75 mm (±0.25). The milled molars reached a mean fracture force of 1817.50 N (±258.22) with a deformation of 1.750 mm (±0.20). The printed incisors fractured at 321.63 N (±145.90) with a deformation of 1.94 mm (±0.40), while the milled incisors fractured at 443.38 N (±113.63) with a deformation of 2.26 mm (±0.40). The milled molar group revealed significantly higher mechanical fracture strength than the 3D-printed molar group (P<0.001).

**Funding:** The author(s) received no specific funding for this work.

**Competing interests:** The authors have declared that no competing interests exist.

However, no significant differences between the 3D-printed incisors and the milled incisors were found (p = 0.084). There was no significant difference in the travel distance until fracture for both the molar group (p = 1.000) and the incisor group (p = 0.129).

## Conclusion

Within the limits of this in vitro investigation, printed and milled temporary crowns withstood masticatory forces and were safe for clinical use.

## Introduction

Temporary crowns and bridges are used for the immediate restoration of prepared teeth. These temporary materials bridge the period until the final restoration is fabricated and placed [1]. Due to modern technology and patient demands, the need for lifelike dental restorations has expanded dramatically in recent years. As a result, the use of diverse innovative restorative materials with high mechanical qualities is critical for both interim and permanent solutions. Accurate temporary restorations are critical because they protect pulpal tissues, prevent bacterial contamination, and preserve periodontal tissues. Furthermore, it is critical to avoid tooth rotation from its natural position in terms of supraocclusion and infraocclusion and to preseve aesthetics and oral functions, including mastication and speaking [2].

In recent years, developments of revolutionary manufacturing technologies, effective restorative materials, and creative clinical procedures have paved the way for digital dentistry [3]. Intraoral scanners (IOSs) and modern computer-aided design/computer-aided manufacturing (CAD/CAM) techniques, such as milling technologies and 3D printing, have enabled the use of novel metal-free dental materials, allowing them to replace standard metal frameworks and to improve the biomimetic and aesthetic results of restorations [4].

Milled resin-composite crowns made using CAD/CAM have emerged as potential alternatives to metallic restorations in recent years [5]. However, some difficulties, such as milling bar degradation, material waste, and stringent requirements for adequate abutment preparation, must be emphasized. Therefore, three-dimensional printing is a promising, rapid, and cost-efficient method for creating dental prostheses digitally. 3D printing is a sophisticated manufacturing technology that uses computer-aided design digital models to automatically generate personalized 3D objects [6]. Ceramics and resin are among the materials that are used in 3D printing. 3D/4D printing can be integrated with artificial intelligence and machine learning for patient-specific medical technology applications [7]

Recent digital techniques for the fabrication of crowns include digital light processing (DLP) and stereolithography (STL), which provide speedy printing and good precision [8]. The DLP approach offers quick printing and excellent accuracy. The item is created according to the CAD design utilizing a resin-filled vat for layer-by-layer photopolymerization on the platform during DLP. Acrylic composites in dental treatment are not new, but 3D printing makes in-office production of intricate parts possible in under 30 minutes, enabling new treatment options [1, 2]. Nevertheless, both conventional and 3D printed materials must fulfill specific mechanical properties to withstand occlusal and masticatory forces [3–12]. To date, 3D-printed materials are used clinically as durable temporary and definitive restorations [13]. The main advantages of CAD/CAM technologies include accuracy, time efficiency, and doability; thus, they are becoming primary health care technologies for solving complex medical

problems, making them promising rapid and economical approaches for the digital fabrication of dental prostheses [7]

CAD/CAM technologies provide superior mechanical strength, excellent esthetic and optical characteristics, and trustworthy precision and accuracy, expanding the clinical spectrum and allowing for novel and less invasive restorative solutions [14]

In terms of digital versus conventional workflows, the benefits lie in digital modeling and virtual planning, allowing several procedures to be performed with software and without human contact. The spread of infectious agents, such as COVID-19, can be more easily limited by reducing the number of work steps and procedures that may generate aerosols and environmental contaminants [15]

The main limitations of CAD/CAM technologies include the high initial cost, the lack of color gradients in 3D-printed prostheses, technology failure, and the learning curve.

To fulfill mechanical and chemical requirements, thorough investigations are mandatory prior to in vivo incorporation. Although clinical use cannot be simulated entirely in vitro with a standardized test [16], it is possible to find material-specific properties in vitro, which are essential for its fundamental understanding. Various methods are available for analyzing the mechanical strengths of materials. One of the most commonly used techniques is compression testing [4–6, 9–11, 13]. Compression tests are conducted by loading the test specimen between two plates and then applying a force to the specimen by moving the crossheads together. During the test, the specimen is compressed, and the deformation and applied load are recorded [9]. This testing is used to assess the material's behavior or reaction under crushing pressures and its plastic flow behavior and ductile fracture limits [10].

As 3D printing resin for temporary crowns has recently been developed, there are limited available data and studies on its mechanical strength [16]. In this study, we aim to analyze the fracture strengths of implant-supported 3D printed temporary crowns relative to milled crowns by compression testing.

We hypothesize that the fracture strengths of the 3D-printed and milled crowns will not show significant differences.

## Materials and methods

In the present investigation, 32 specimens were digitally designed using Aadva Software (GC, Tokyo, Japan) (Figs 1 and 2) by estimating the effect size 0.5 with 80% power (alpha = .05, two-tailed). The 3D printer Varseo S (Bego, Bremen, Germany) and the milling unit inLab MC X5 (Sirona, Bensheim, Germany) were used to produce 16 specimens per group from the same digitally designed crowns (Fig 3). Each group consisted of eight maxillary central incisor crowns (tooth 11) and eight maxillary molar crowns (tooth 16).

DLP (digital light processing) is an additive technology that was applied for the printed specimens. For 3D printing, a mask printer (Varseo, BEGO, Bremen, Germany) and temporary material (VarseoSmile Temp A3, BEGO, Bremen, Germany) were used. The resin was a light-curing, free-flowing plastic based on methacrylic acid esters for 3D printing.

For the milled crowns, a millable temporary material was used (VitaCAD Temp monocolor, Vita, Bad Säckingen, Germany). The material consisted of a composite made of a highly cross-linked acrylate polymer with a microfiller for milling.

After pretesting for the fitting of both groups, the cement gap was evaluated. The cement gap in the printed group was set digitally at 40 μm. In comparison, the gap was 60 μm in the milled groups to ensure maximum retention [17].

For the 3D printed samples, supports were placed on the occlusal surfaces and incisal edges away from the screw path. Thus, uniform cementation surfaces were guaranteed without

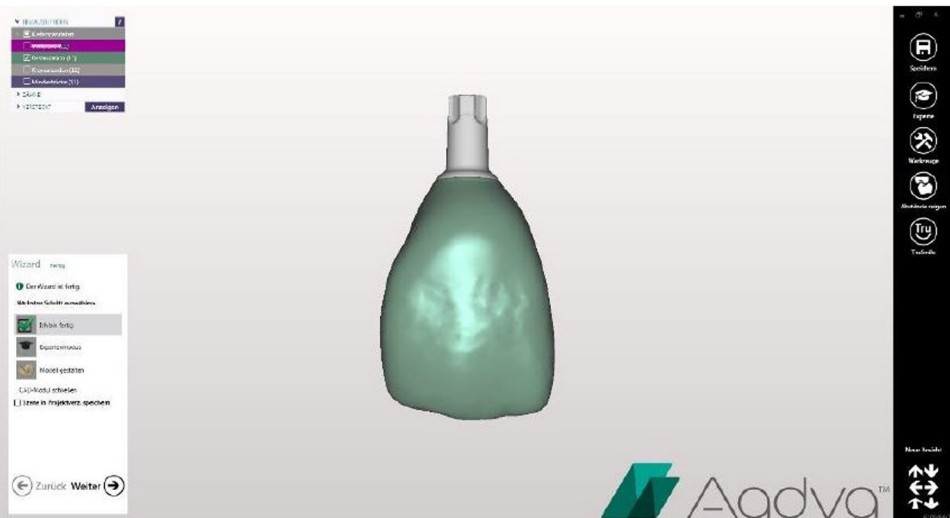

**Fig 1. Virtual planning of tooth crown 11.**

abrasive reworking. Their postprocessing was performed according to the manufacturer's instructions. In summary, an ultrasonic 96% ethanol bath was used to clean the specimens. Then, the specimens were dried with compressed air. Next, the surface was cured using the polymerization unit Otoflash with two cycles of 1500 flashes (Bego, Bremen, Germany).

The crowns were mounted on Ti-base SICvantage CAD/CAM straight implants with an abutment angle of 15° for incisors and 0° for molars (SIC invent AG, Basel, Switzerland).

The test samples were loaded into the universal testing machine z010 (Zwick/Roell, Ulm, Germany).

In order from first to last, we measured the fracture forces between the molar groups, the fracture forces between the incisor groups, the deformations between the molar groups and the deformations between the incisor groups. Thus, it was possible to determine whether the

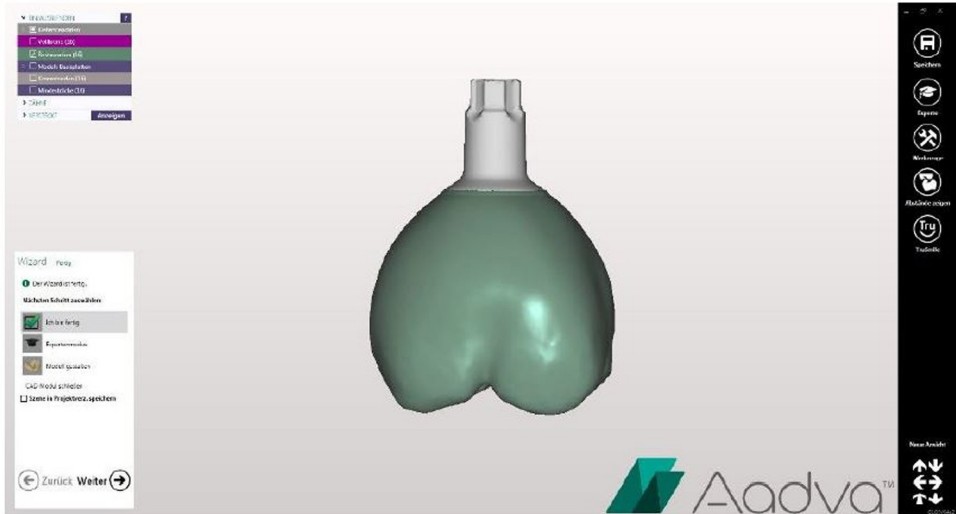

**Fig 2. Virtual planning of tooth crown 16.**

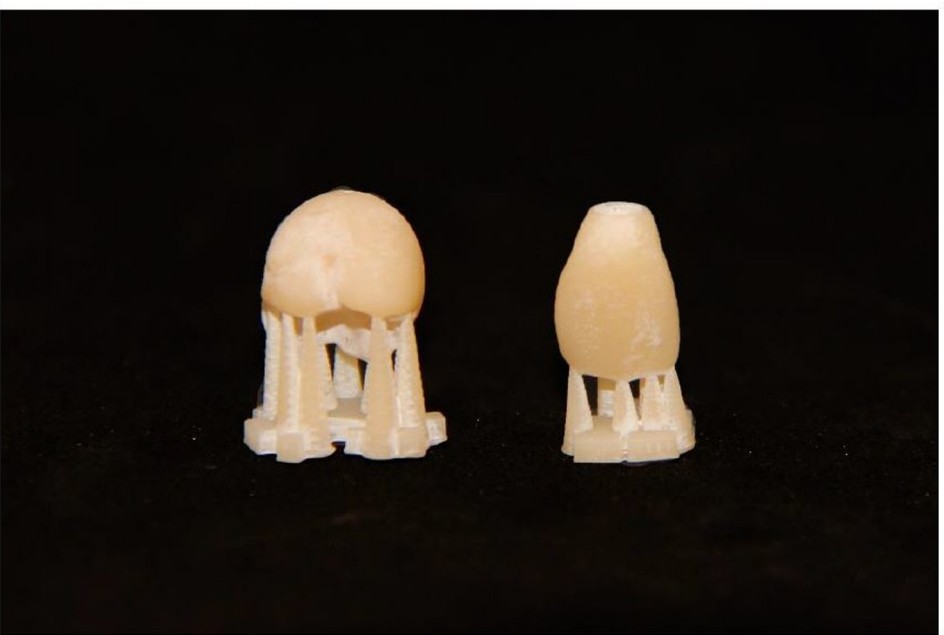

**Fig 3. 3D printing of provisional crowns with support structures.**

materials differed in fracture resistance or deformation and whether there were differences within tooth regions.

## Measurement of the fracture strength

For fracture resistance testing, a load was applied at a cross-head speed of 0.5 mm/min and at the tooth until fracture occurred, according to ISO 11405/2003. The molar specimens were loaded along the implant axis and the incisors at an angle of 45˚ to simulate the natural teeth position and protrusion forces (Figs 1–3) [18].

The loading forces were recorded in Newtons [N], while the travel distance until failure was measured in millimeters [mm].

## Statistical analysis

The statistical analysis was performed using SPSS Version 24.0 (International Business Machines Corporation, New York, USA). A Shapiro–Wilk test was applied to test the normal distribution of the data, and t tests were performed to compare the two groups. The statistical significance was set at $p<0.05$.

## Results

The test data were normally distributed in all groups. The mean values and standard deviations (M±SD) of the maximum forces and the travel distance for all groups are shown in Table 1.

The mean fracture strength of the printed molars was 1189.50 N (±250.85) with a deformation of 1.75 mm (±0.25), whereas the milled molars reached a mean fracture strength of 1817.50 N (±258.22) with a deformation of 1.75 mm (±0.20). The printed incisors fractured at 321.63 N (±145.90) with a deformation of 1.94 mm (±0.40), while the milled incisors fractured at 443.38 N (±113.63) with a deformation of 2.26 mm (±0.40) (Figs 4–8).

Table 1. Descriptive statistics for all groups.

| Group | | Maximum Forces (N) | Travel Distance (mm) |
|---|---|---|---|
| 3D Printed 11 | Mean | 324.38 | 1.938 |
| | SD | 145.010 | 0.4033 |
| Milled 11 | Mean | 451.13 | 2.263 |
| | SD | 111.716 | 0.4033 |
| 3D Printed 16 | Mean | 1203.38 | 1.750 |
| | SD | 251.861 | 0.2449 |
| Milled 16 | Mean | 1850.00 | 1.750 |
| | SD | 253.659 | 0.2000 |

The milled molar group revealed significantly higher mechanical fracture strength than the 3D printed molar group (p<0.001). However, no significant differences between the 3D printed incisors and the milled incisors were found (p = 0.084). There were no significant differences in the travel distances until fracture for both the molar group (p = 1.000) and the incisor group (p = 0.129).

All crowns exhibited similar brittle fracture patterns with audible crackling beforehand. After the propagation of an initial occlusal crack, a sudden fracture occurred. The fracture extended downward from the occlusal surface and divided the crown in the middle through the inlet for the abutment.

## Discussion

The purpose of this study is to investigate the mechanical performance of 3D-printed and milled temporary crowns. The significance of exact preliminary restorations is widely acknowledged. Furthermore, there is some indication that 3D-printed provisional restorations may be preferable to their CAD/CAM counterparts [19]. The development of provisional 3D-printed restorations promises a simpler manufacturing method and possibly stronger provisional restorations than conventional techniques. Therefore, we compare the fracture resistance levels of 3D printed and milled temporary crowns by compression testing on dental implants in vitro.

The findings show significantly higher fracture strengths for milled molars than for 3D printed molars (p<0.001) but no significant differences in the incisor crowns (p = 0.084). In contrast, the differences in travel distances until failure are not significant for either molars (p = 1.000) or incisors (p = 0.129).

The results in the present study are in line with previously published articles. The meta-analysis of Jain et al. features articles comparing the physical and mechanical properties of 3D-printed provisional crown and FDP resin materials with CAD/CAM milled and conventional provisional resins. The researchers conclude that 3D-printed provisional crown and FDP resin materials have inferior physical properties relative to CAD/CAM milled materials [19]. Furthermore, Ellakany et al. have investigated the mechanical properties of CAD/CAM milled and two different types of 3D-printed, 3-unit IFDPs relative to conventional IFDPs after the thermomechanical aging process. The conclusion of the study is that superior flexural strength, elastic modulus, and hardness are reported for milled IFDPs [20].

If both test groups are considered from a mechanical point of view, the force acting on the molars is perpendicular, which is in contrast to the incisors, where the force acts at an angle of 45˚ and the tooth surface is negligible. In this case, the incisors are loaded differently than the molars since the crown (force arm) and the acting force form a cross product, which acts as a

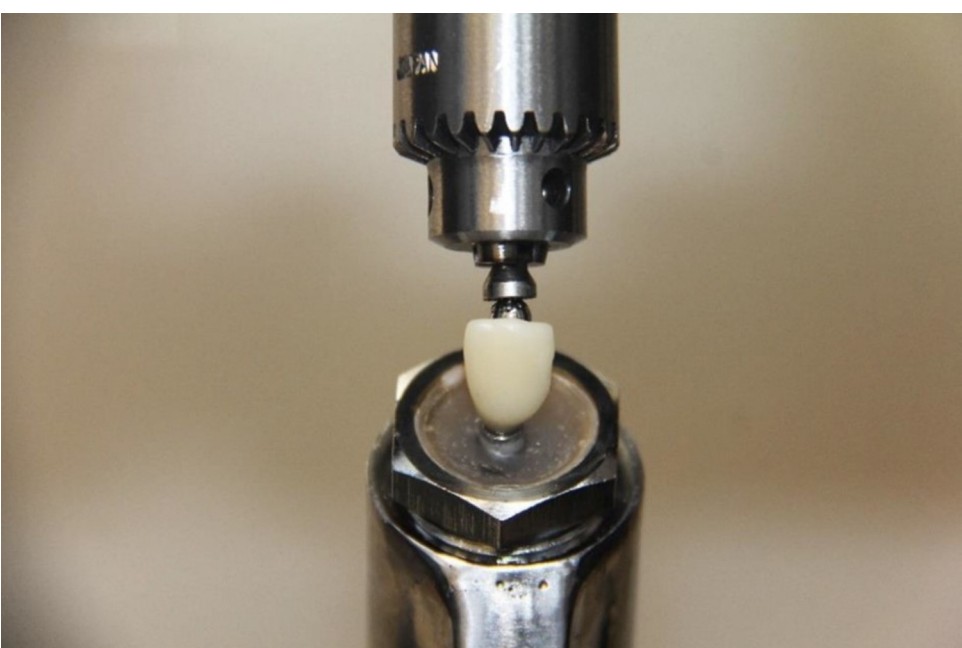

**Fig 4. Main testing for provisional crown 11.**

torque on the rotational axis of the crown. Due to this lever, a greater force acts mechanically on the crown than is indicated on the machine. In the case of the molars, there is no torque because the force acts perpendicularly on the crown.

Screw retained abutments are chosen to ensure consistent test results and reproducibility while remaining true to the clinical reality [21]. The fracture resistance test is chosen because it

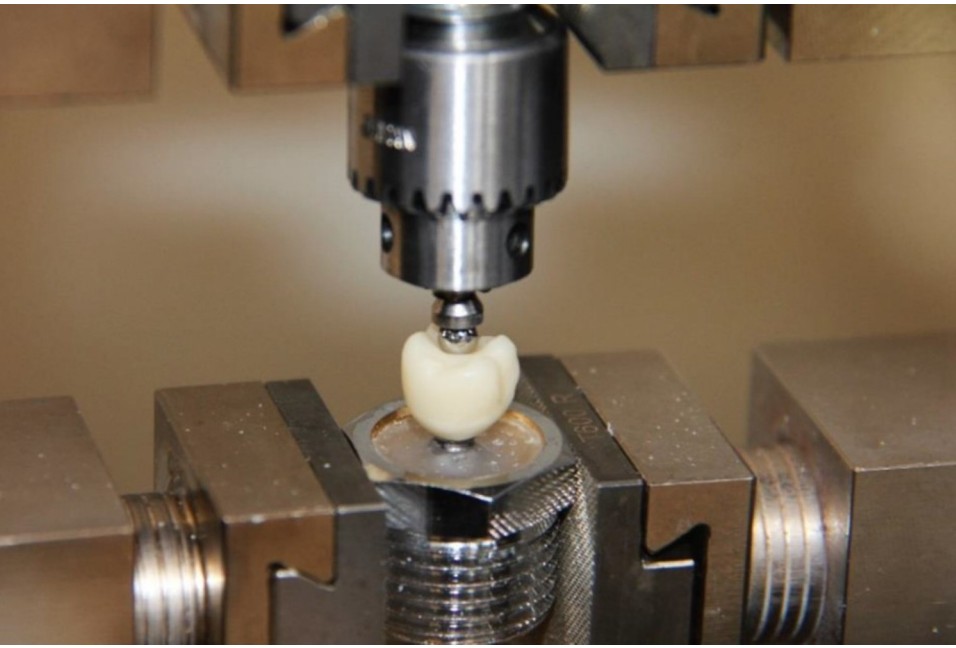

**Fig 5. Main testing for provisional crown 16.**

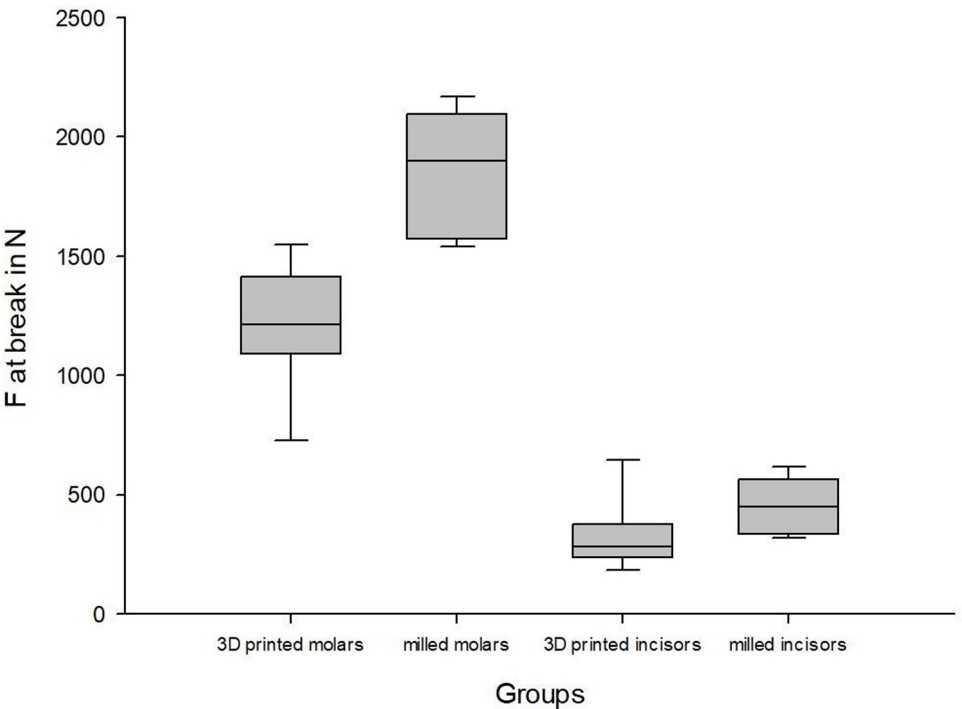

**Fig 6. Boxplot force at break.**

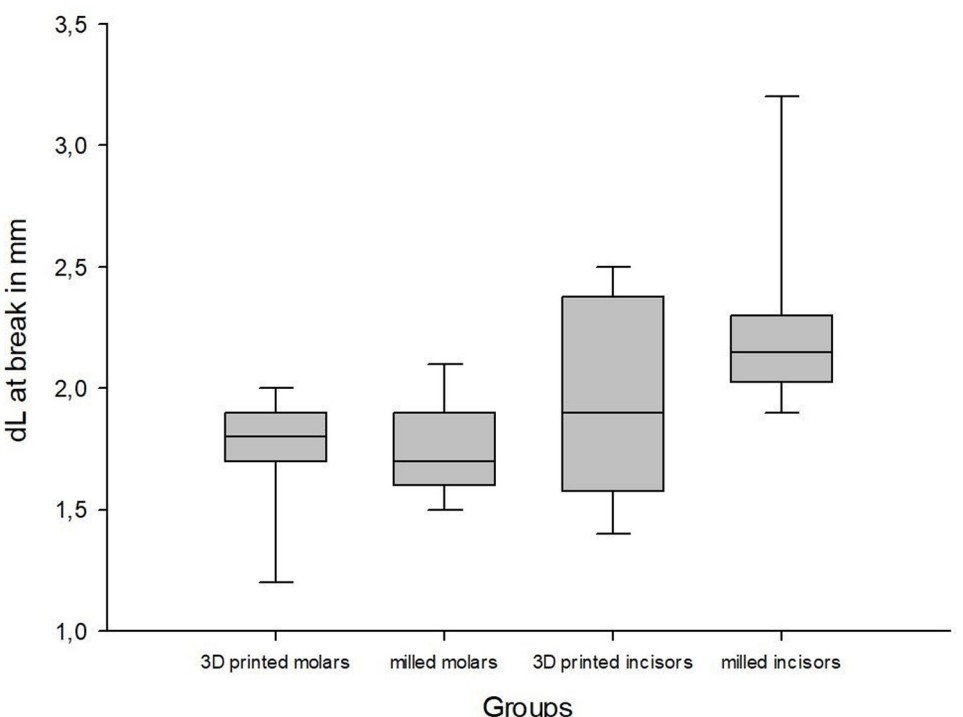

**Fig 7. Boxplot dL at break.**

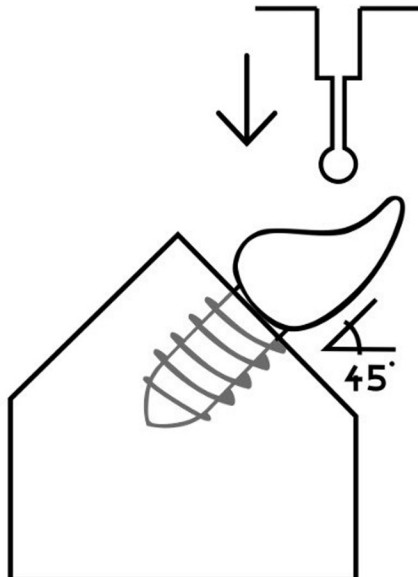 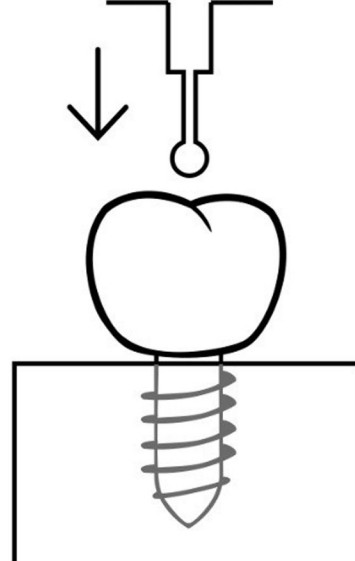

**Fig 8. Testing setup.**

is a common and standardized test that can be easily compared to the literature [4–6, 9–13]. This test provides comparable and straightforward results. Crown designs are kept consistent within the groups. In vivo, crown designs and layer thickness will differ from this study. Nevertheless, consistent dimensions in vitro are essential for ensuring comparable results of material-specific properties. Although it is not as crucial for permanent restorations, a drawback of the fracture test is that it does not simulate long-term intraoral wear.

Different studies are performed with various crown designs or testing setups [4–6, 9–13]. Consequently, it is difficult to compare the specific fracture values of the milled and 3D printed crowns obtained in the present study to those of other similar publications. For this reason, the resulting values are compared inside the present study. Nonetheless, the relationships of the fracture strengths between materials are compared. In the literature, fracture values for temporary crowns are found, with PMMA and composite-based materials usually having higher values than hand-manufactured chair-side temporary restorations. A study conducted by Karaokutan I. et al. shows that composite-based materials have the highest fracture strength, while conventional chairside temporary restorations have the lowest values [5]. Consistent with the present investigation, a study conducted by Reeponmaha T. et al. shows higher variances in fracture values for 3D-printed crowns than for milled crowns [22]. Further research is needed to provide insight into this matter. Inconsistency and errors during polymerization or differences in positioning between samples on the print platform may lead to higher variances in 3D-printed crowns than in milled crowns [23].

All crowns follow similar brittle fracture patterns with audible crackling beforehand. After the propagation of an initial occlusal crack, a sudden fracture occurs. The fracture extends downward from the occlusal surface and divides the crown in the middle through the inlet for the abutment. Different crown designs and layer thicknesses may change the location and direction of fracture [24].

Very little data exist in the literature regarding the fracture strengths of these modern provisional materials. The results support that the fracture strengths of DLP-based 3D-printed

incisor crowns are similar to those of milled crowns. In contrast, milled molar crowns show significantly higher fracture strengths than 3D printed crowns.

The fact that just one system for each manufacturing process is studied is one of the study's limitations. Furthermore, precision as a criterion of crown accuracy across different production procedures needs further research. DLP-based 3D printing demonstrates good accuracy for the mediodistal and buccolingual widths of dental models. However, the discrepancies in accuracy may be due to thickness and layer shrinking, particularly in the Z-axis. Therefore, further studies are needed to establish guidelines for optimum dimensions for each application while using different manufacturing processes and printing conditions.

Nonetheless, both printed and milled temporary crowns withstand masticatory forces and are safe for clinical use [25]. Further research on 3D-printed materials is needed to determine other chemical and physical properties, such as coloring, during intraoral use. Temporary CAD/CAM restorations allow the clinician to choose an emergence profile that stabilizes the gingiva and prevents severe shrinkage after extractions [26]. The situation worked out in the planning phase can be transferred to the temporary situation. Additionally, aesthetics and function can be tested by the patient.

## Conclusion

Within the limitations of the present study, the fracture strength testing shows that milled molar crowns (1817.50 N ±258.22) have significantly higher fracture strengths than 3D printed crowns (1189.50 N ±250.85); however, for incisor crowns, no statistically significant differences are shown in the present study (443.38 N ±113.63 vs. 321.63 N ±145.90). In all cases, for both crown shape and production method, no statistically significant differences are apparent in terms of deformation.

Regarding the results from a clinical point of view, the significant differences between the crowns in terms of fracture strength withstand masticatory forces, making them safe for clinical use.

### Clinical significance

3D-printed crowns are considered an alternative for long-term provisional prosthetic restorations.

### Supporting information

**S1 File.**
(PDF)

## Acknowledgments

The authors acknowledge Dr.med.dent Moritz Wagner for his scientific support.

## Author Contributions

**Conceptualization:** Ahmed Othman.

**Data curation:** Ahmed Othman.

**Formal analysis:** Maximillian Sandmair.

**Methodology:** Ahmed Othman, Maximillian Sandmair.

**Project administration:** Ahmed Othman.

**Resources:** Ahmed Othman.

**Software:** Maximillian Sandmair.

**Supervision:** Constantin von See.

**Validation:** Ahmed Othman.

**Visualization:** Ahmed Othman.

**Writing – original draft:** Ahmed Othman.

**Writing – review & editing:** Vasilios Alevizakos.

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
