## [Decision Letter · Decision Letter 0]

24 Nov 2022

PONE-D-22-19019Comparative mechanical testing of 3D-printed versus milled provisional crowns: An in vitro studyPLOS ONE

Dear Dr. Othman,

Thank you for submitting your manuscript to PLOS ONE. After careful consideration, we feel that it has merit but does not fully meet PLOS ONE’s publication criteria as it currently stands. Therefore, we invite you to submit a revised version of the manuscript that addresses the points raised during the review process.

Thank you for submitting this interesting manuscript. Although the reviewers were interested in the topic, they have raised substantial concerns related to the study design and presentation. Please address the reviewers' comments and provide suitable rebuttal/s if you are unable to do so. I look forward to receiving the revised manuscript.

We look forward to receiving your revised manuscript.

Kind regards,

Mirza Rustum Baig

Academic Editor

PLOS ONE

Journal Requirements:

2. Please amend your list of authors on the manuscript to ensure that each author is linked to an affiliation. Authors’ affiliations should reflect the institution where the work was done (if authors moved subsequently, you can also list the new affiliation stating “current affiliation:….” as necessary).

Reviewers' comments:

Reviewer's Responses to Questions

**Comments to the Author**

1. Is the manuscript technically sound, and do the data support the conclusions?

Reviewer #1: Yes

Reviewer #2: No

Reviewer #3: Partly

2. Has the statistical analysis been performed appropriately and rigorously? 

Reviewer #1: Yes

Reviewer #2: Yes

Reviewer #3: I Don't Know

3. Have the authors made all data underlying the findings in their manuscript fully available?

Reviewer #1: Yes

Reviewer #2: Yes

Reviewer #3: No

4. Is the manuscript presented in an intelligible fashion and written in standard English?

Reviewer #1: Yes

Reviewer #2: Yes

Reviewer #3: No

5. Review Comments to the Author

Reviewer #1: This study aims to analyze the fracture resistance of implant-supported 3D printed temporary crowns compared to milled crowns by compression testing. And they found that the milled molar group revealed significantly higher mechanical fracture strength than the 3D printed molar group (P<0.001). However, no significant difference between the 3D printed incisors and the milled incisors was found (P=0.084). Some comments.

Title. It is better to replace the “Comparative mechanical testing” in title with more specific testing.

Abstract. If possible, it is better to add brief on the printed and milling technique used. Which materials were used for printing implant crowns.

Introduction.

The introduction is short. It will be better to add details on the Digital techniques of fabrication of crowns. Then explaining more on the 3D printing and milling technologies with indications, advantages and limitations. Some recent literatures are.

https://link.springer.com/chapter/10.1007/978-981-16-7152-4_12

https://www.ncbi.nlm.nih.gov/pmc/articles/PMC7356564/

Method.

Please add on the simple size calculation.

Please discuss more on the results differences between the molars and incisors.

Reviewer #2: Dear Authors,

The aim of this study was to analyze the fracture resistance of implant-supported 3D printed temporary crowns compared to milled crowns by compression testing. As the full-length article, the Introduction section and the Discussion section were too short. Only fracture loads are not enough. Fatigue test and finite element analysis should be contained to be strengthen this manuscript.

Reviewer #3: The abstract need revision. Materials and Methods need to be rewritten again. Discussion need to include comparisons with similar previously published articles. References are not as per journal requirements. Sample size is not justified and very small. What is the power of this study ???. What kind of Force was applied on incisor and molars , how much ???. Conclusion should be revised. What is the mean fracture strength between the group of crowns for incisors and molars ???

6. PLOS authors have the option to publish the peer review history of their article (what does this mean?). If published, this will include your full peer review and any attached files.

Reviewer #1: **Yes: **Dinesh Rokaya

Reviewer #2: No

Reviewer #3: No

---

## [Author Response · Author response to Decision Letter 0]

21 Dec 2022

December 20, 2022

Dear Dr. Mirza Rustum Baig, 

Thank you for giving me the opportunity to submit a revised draft of my manuscript titled “Comparative mechanical testing of 3D-printed versus milled provisional crowns: An in vitro study" to the Journal PLOS ONE. We appreciate the time and effort you and the reviewers have dedicated to providing valuable feedback on the manuscript, responded appropriately as noted below, and highlighted the changes within the manuscript.

Reviewer 1:

This study aims to analyze the fracture resistance of implant-supported 3D-printed temporary crowns compared to milled crowns by compression testing. And they found that the milled molar group revealed significantly higher mechanical fracture strength than the 3D-printed molar group (P<0.001). However, no significant difference between the 3D-printed incisors and the milled incisors was found (P=0.084). Some comments.

1. Title. It is better to replace the “Comparative mechanical testing” in the title with more specific testing.

Response: Thank you for your guidance and advice. Done

“The fracture resistance of 3D-printed versus milled provisional crowns: An in vitro study”

2. Abstract. If possible, it is better to add brief on the printed and milling technique used. Which materials were used for printing implant crowns?

Response: Done 

“The first group (16 specimens) was 3D printed by a mask printer (Varseo, BEGO, Bremen, Germany) with a temporary material (VarseoSmile Temp A3, BEGO, Bremen, Germany). The second group was milled in which a millable temporary material was used (VitaCAD Temp monocolor, Vita, Bad Säckingen, Germany).”

3. Introduction. The introduction is short. It will be better to add details on the Digital techniques of fabrication of crowns. Then explaining more on the 3D printing and milling technologies with indications, advantages and limitations. Some recent literatures are. https://link.springer.com/chapter/10.1007/978-981-16-7152-4_12

https://www.ncbi.nlm.nih.gov/pmc/articles/PMC7356564/

Response: Done

“Milled resin-composite crowns made using CAD/CAM have emerged as a potential alternative to metallic restorations in recent years [5]. However, some difficulties such as milling bar degradation, material waste, and stringent requirements for adequate abutment preparation must be emphasized. Therefore, three-dimensional printing is a promising, rapid, and cost-efficient method of creating dental prostheses digitally. It is a sophisticated manufacturing technology that use computer-aided design digital models to automatically generate personalized 3D objects [6]. Ceramics and resin are among the materials that be used in 3D printing. Digital light processing (DLP) and stereolithography (STL) provide speedy printing and good precision [7]. The DLP approach offers quick printing and excellent accuracy. The item is created according to the CAD design utilizing a resin-filled vat for layer-by-layer photopolymerization on the platform in the DLP process.”

4. Method. Please add on the sample size calculation.

Response: Done

“In the present investigation, 32 specimens were digitally designed using Aadva Software (GC, Tokyo, Japan) by estimation of the effect size 0.5 with 80% power (alpha = .05, two-tailed).”

5. Please discuss more on the results differences between the molars and incisors.

Response: Done

“If both test groups are considered from a mechanical point of view, the force acting on the molars is perpendicular, in contrast to the incisors, where the force acts at an angle of 45° - the tooth surface is negligible. In this case, the incisors are loaded differently than the molars, since the crown (force arm) and the acting force form a cross product, which acts as a torque on the rotational axis of the crown. Due to this lever, a greater force acts mechanically on the crown than is indicated on the machine. In the case of the molars, there is no torque because the force acts perpendicularly on the crown.”

Reviewer 2:

The aim of this study was to analyze the fracture resistance of implant-supported 3D-printed temporary crowns compared to milled crowns by compression testing.

1. As the full-length article, the Introduction section and the Discussion section were too short.

Response: Done

2. Only fracture loads are not enough. 

Response: Thank you for your comment. We agree with you. We used the compression test which is One of the most used techniques. It is used to assess a material's behavior or reaction under crushing pressures, as well as to test a material's plastic flow behavior and ductile fracture limits. Compression tests are conducted by loading the test specimen between two plates and then applying a force to the specimen by moving the crossheads together. During the test, the specimen is compressed, and deformation versus the applied load is recorded. This study aims to analyze the fracture strength of implant-supported 3D-printed temporary crowns compared to milled crowns by compression testing. That would be addressed by measuring; first the fracture forces between the molar groups, Secondly the fracture forces between the incisor groups, Thirdly the deformation between the molar groups, and lastly the deformation between the incisor groups. Thus, it was possible to determine whether the materials differed in fracture resistance or deformation and whether there were differences within tooth regions. We hypothesized that the fracture strength of the 3D-printing method is non-inferior to that of the milling method.

3. Fatigue test and finite element analysis should be contained to strengthen this manuscript.

Response: Thank you for your comment. We agree, a finite element analysis would strengthen this manuscript. But also, without a finite element analysis reader can have clinically relevant and useful information from our manuscript.

Reviewer 3:

1. The abstract need revision. 

Response: Sorry for the inconvenience. Done

“Abstract

Aims: This study aims to analyze the fracture resistance of implant-supported 3D-printed temporary crowns compared to milled crowns by compression testing.

Methods: The study sample included 32 specimens of temporary crowns which were divided into 16 specimens per group. Each group consisted of eight maxillary central incisor crowns (tooth 11) and eight maxillary molar crowns (tooth 16). The first group (16 specimens) was 3D printed by a mask printer (Varseo, BEGO, Bremen, Germany) with a temporary material (VarseoSmile Temp A3, BEGO, Bremen, Germany). The second group was milled in which a millable temporary material was used (VitaCAD Temp mono-color, Vita, Bad Säckingen, Germany). The two groups were compression tested until failure to estimate their fracture resistance. The loading forces and travel distance until failure were measured. The statistical analysis was performed using SPSS Version 24.0. We performed multiple t-tests and considered a significance level of p <0.05.

Results: The mean fracture force of the printed molars was 1189.50N (±250.85) with a deformation of 1.75mm (±0.25), whereas the milled molars reached 1817.50N (±258.22) with a deformation of 1.750mm (±0.20). The printed incisors fractured at 321.63N (±145.90) with a deformation of 1.94mm (±0.40) while the milled ones fractured at 443.38N (±113.63) with a deformation of 2.26mm (±0.40). The milled molar group revealed significantly higher mechanical fracture strength than the 3D-printed molar group (P<0.001). However, no significant difference between the 3D-printed incisors and the milled incisors was found (p=0.084). There was no significant difference in the travel distance until fracture for both the molar group (p=1.000) and the incisors (p=0.129). Conclusion: Within the limits of this in-vitro investigation, the printed and milled temporary crowns can both withstand the masticatory forces and are safe for clinical use.”

2. Materials and Methods need to be rewritten again. 

Response: Done

3. Discussion need to include comparisons with similar previously published articles.

Response: Done

“The results in the present study are in line with previously published articles. The meta-analysis of Jain et al. compared articles comparing the physical and mechanical properties of 3D-printed provisional crown and FDP resin materials with CAD/CAM milled and conventional provisional resins. They concluded that that 3D-printed provisional crown and FDP resin materials have inferior physical properties compared to CAD/CAM milled ones (23). Furthermore, Ellakany et al. investigated the mechanical properties of CAD/CAM milled and two different types of 3D-printed, 3-unit IFDPs in comparison to the conventional IFDPs after the thermo-mechanical aging process. The conclusion of the study was that superior flexural strength, elastic modulus, and hardness were reported for milled IFDPs (24).”

4. References are not as per journal requirements. 

Response: Thank you for the kind information. References style and format were checked to be following journal guidelines.

5. Sample size is not justified and very small. What is the power of this study???. 

Response: thank you for your comment. In the present investigation, 32 specimens were digitally designed using Aadva Software (GC, Tokyo, Japan) by estimation of the effect size 0.5 with 80% power (alpha = .05, two-tailed). The 3D printer Varseo S (Bego, Bremen, Germany) and the milling unit inLab MC X5 (Sirona, Bensheim, Germany) were used to produce 16 specimens per group from the same digitally designed crowns. Each group consisted of eight maxillary central incisor crowns (tooth 11) and eight maxillary molar crowns (tooth 16).

6. What kind of Force was applied on incisor and molars, how much ???. 

Response: Done

“For fracture resistance testing a load was applied at a cross-head speed of 0.5 mm/min and at the tooth until fracture occurred, according to ISO 11405/2003 (21).”

7. Conclusion should be revised. What is the mean fracture strength between the group of crowns for incisors and molars???

Response: Done

“Within the limitations of the present study, the fracture strength testing showed that milled molars crowns (1817.50N ±258.22) have statistically significantly higher fracture strength compared to 3D printed ones (1189.50N ±250.85), but in case of incisor crowns no statistically significant difference was shown in the present study (443.38N ±113.63 vs. 321.63N ±145.90). In all cases either crown shape and production method no statistically significant differences were shown in terms of deformation.

Regarding the results from a clinical point of view, those significant differences between the crowns in terms of fracture strength both can withstand the masticatory forces and are safe for clinical use.”

---

## [Decision Letter · Decision Letter 1]

16 Jan 2023

PONE-D-22-19019R1The fracture resistance of 3D-printed versus milled provisional crowns: An in vitro studyPLOS ONE

Dear Dr. Othman,

Thank you for submitting your manuscript to PLOS ONE. After careful consideration, we feel that it has merit but does not fully meet PLOS ONE’s publication criteria as it currently stands. Therefore, we invite you to submit a revised version of the manuscript that addresses the points raised during the review process.

We look forward to receiving your revised manuscript.

Kind regards,

Mirza Rustum Baig

Academic Editor

PLOS ONE

Reviewers' comments:

Reviewer's Responses to Questions

**Comments to the Author**

1. If the authors have adequately addressed your comments raised in a previous round of review and you feel that this manuscript is now acceptable for publication, you may indicate that here to bypass the “Comments to the Author” section, enter your conflict of interest statement in the “Confidential to Editor” section, and submit your "Accept" recommendation.

Reviewer #1: (No Response)

Reviewer #3: All comments have been addressed

Reviewer #4: All comments have been addressed

2. Is the manuscript technically sound, and do the data support the conclusions?

Reviewer #1: Yes

Reviewer #3: Partly

Reviewer #4: Partly

3. Has the statistical analysis been performed appropriately and rigorously? 

Reviewer #1: Yes

Reviewer #3: I Don't Know

Reviewer #4: Yes

4. Have the authors made all data underlying the findings in their manuscript fully available?

Reviewer #1: No

Reviewer #3: No

Reviewer #4: (No Response)

5. Is the manuscript presented in an intelligible fashion and written in standard English?

Reviewer #1: Yes

Reviewer #3: No

Reviewer #4: Yes

6. Review Comments to the Author

Reviewer #1: The authors have not addressed some previous comments.

Introduction. It will be better to add details on the Digital

techniques of fabrication of crowns. Then explaining more on the 3D printing and milling

technologies with indications, advantages and limitations. Some recent literatures are.

https://link.springer.com/chapter/10.1007/978-981-16-7152-4_12

https://www.ncbi.nlm.nih.gov/pmc/articles/PMC7356564/

Reviewer #3: Minor revision includes English language editing, providing complete statistics for assessment.kindly check image quality including resolution, pixels , blurring if any . I would appreciate more pictures from groups. Please add on clinical significance of this study . From my side as of now my queries had been met.

Reviewer #4: Dear Authors,

The topic is interesting but unfortunatelly there are some important lacking. For example; The abstract is poor and not informative. The test groups are inadequate. Because there are different materials for this purpose. Are the crowns include screw hole ore not? How does effect this feature to study? Generally, the information presented are not new, and does not add anything new to the current understanding. This study can't serve important clinical results.

7. PLOS authors have the option to publish the peer review history of their article (what does this mean?). If published, this will include your full peer review and any attached files.

Reviewer #1: No

Reviewer #3: **Yes: **Prof Dr Hariharan Ramakrishnan

Reviewer #4: No

---

## [Author Response · Author response to Decision Letter 1]

21 Feb 2023

15.Feb.2023

Dear Dr. Mirza Rustum Baig, 

Thank you for giving me the opportunity to submit a revised draft of my manuscript titled “The fracture resistance of 3D-printed versus milled provisional crowns: An in vitro study" to the Journal PLOS ONE. We appreciate the time and effort you and the reviewers have dedicated to providing valuable feedback on the manuscript, responded appropriately as noted below, and highlighted the changes within the manuscript.

Reviewer 1:

The authors have not addressed some previous comments.

Introduction. It will be better to add details on the Digital techniques of fabrication of crowns. Then explaining more on the 3D printing and milling technologies with indications, advantages and limitations. Some recent literatures are:

https://link.springer.com/chapter/10.1007/978-981-16-7152-4_12

https://www.ncbi.nlm.nih.gov/pmc/articles/PMC7356564/

1. The authors have not addressed some previous comments.

Response: We would like to apologize for any inconvenience. The missing comments have been precisely addressed as much as possible. Thank you for your guidance, advice and understanding.

2. Introduction. It will be better to add details on the Digital techniques of fabrication of crowns. Then explaining more on the 3D printing and milling technologies with indications, advantages and limitations. 

Response: Done 

“3D/4D printing can be integrated with artificial intelligence and machine learning to apply for patient-specific medical technologies (7). The recent digital techniques of fabrication of crowns includes Digital light processing (DLP) and stereolithography (STL) which provide speedy printing and good precision (8). The main advantages of CAD/CAM technologies includes accuracy, time-efficieny, doability and becoming a main part of healthcare technology solving complex medical problems which are promising for a rapid and economical technology for the digital fabrication of dental prostheses (7) They also provide superior mechanical strength, excellent esthetic and optical characteristics, and trustworthy precision and accuracy, which expand the clinical spectrum and allow for novel and less invasive restorative solutions (14). In terms of digital versus conventional workflow, the benefit lies in digital modeling and virtual planning, allowing several procedures to be performed with software and without human contact. The spread of infectious agents as COVID-19 can be more easily limited by reducing the number of work steps and procedures that can generate aerosols and environmental contaminants (15). The main limitations of CAD/CAM technologies include the high initial cost, the lack of color gradients in 3D-printed prostheses, technology failure, and the learning curve.

Reviewer 3:

Minor revision includes English language editing, providing complete statistics for assessment. kindly check image quality including resolution, pixels , blurring if any . I would appreciate more pictures from groups. Please add on clinical significance of this study. From my side as of now my queries had been met.

1. Minor revision includes English language editing, providing complete statistics for assessment.

Response: The English language editing was controlled by Springer Nature; certificate is attached for your preference. The statistics analysis for your kind assessment are attached as supplementary file.

2. kindly check image quality including resolution, pixels, blurring if any

Response: Done following Preflight Analysis and Conversion Engine (PACE) digital diagnostic tool, https://pacev2.apexcovantage.com/.

3. I would appreciate more pictures from groups.

Response: Done. More pictures were added in the article (Fig.1-3). 

4. Please add on clinical significance of this study.

Response: Done. clinical significance was added in the article.

5. From my side as of now my queries had been met.

Response: Thank you for the kind understanding, advice and effort. 

Reviewer 4:

The topic is interesting but unfortunately there are some important lacking. For example; The abstract is poor and not informative. The test groups are inadequate. Because there are different materials for this purpose. Are the crowns include screw hole or not? How does effect this feature to study? Generally, the information presented are not new, and does not add anything new to the current understanding. This study can't serve important clinical results.

1. The topic is interesting but unfortunately there are some important lacking. For example; The abstract is poor and not informative.

 Response: Done. The abstract was modified to be more informative.

2. The test groups are inadequate. Because there are different materials for this purpose.

Response: Both milled and 3D printed materials must fulfill specific mechanical properties to withstand the occlusal masticatory forces and can be used clinically as durable temporary restorations as well as possibility for definitive restorations production. Both materials are prescribed as long-term temporary which can withstand mechanical forces. Accordingly, testing both materials for evaluating the comparative forces is considered reasonable. Especially that milled crowns are considered the gold-standard. Comparing printed material to the bench-mark milled material might be considered adequate. 

3. Are the crowns include screw hole or not? How does effect this feature to study?

Response: They don’t include screw hole. The spacer thicknesses were between 30 and 50 μm were tested for both the anterior and posterior regions. The spacer settings of 40 μm ultimately produced the best results on the abutment, whereupon these were adopted for the anterior and posterior region. However, since for the milled temporaries the default standard settings in the Exocad software of 60 μm had been defined and the findings obtained for the printed temporaries could not be transferred to the milled temporaries. the two materials had to be produced with different spacer thicknesses. Current studies showed that in both 3D printing and milling, the difference between digital milling between the digital setting and the real result are sometimes enormous. differences occur in spacer thicknesses. (Hoang, Lisa N.; Thompson, Geoffrey A.; Cho, Seok-Hwan; Berzins, David W.; Ahn, Kwang Woo (2015): The spacer thickness reproduction for central incisor crown fabrication with combined computer-aided design and 3D printing technology: an in vitro study. In: The Journal of prosthetic dentistry 113 (5), S.398–404. DOI: 10.1016/j.prosdent.2014.11.004).

4. Generally, the information presented are not new, and does not add anything new to the current understanding. This study can't serve important clinical results.

Response: clinical use cannot be simulated entirely in vitro with a standardized test, it is possible to find material-specific properties in vitro, which are essential for its fundamental understanding. Various methods are available to analyze the mechanical strength of materials. One of the most used techniques is compression testing.

---

## [Editor Report · Decision Letter 2]

27 Mar 2023

PONE-D-22-19019R2The fracture resistance of 3D-printed versus milled provisional crowns: An in vitro studyPLOS ONE

Dear Dr. Othman,

Thank you for submitting your manuscript to PLOS ONE. After careful consideration, we feel that it has merit but does not fully meet PLOS ONE’s publication criteria as it currently stands. Therefore, we invite you to submit a revised version of the manuscript that addresses the points raised during the review process.

Thank you for submitting the revised manuscript with the responses to the reviewers' comments. Please elaborate on the limitations of the study in the 'Discussion' section, including lack thereof fatigue loading tests (chewing simulation) with thermocycling prior to the fracture resistance tests.  ==============================

We look forward to receiving your revised manuscript.

Kind regards,

Mirza Rustum Baig

Academic Editor

PLOS ONE
---

## [Author Response · Author response to Decision Letter 2]

29 Mar 2023

28.March.2023

Dear Dr. Mirza Rustum Baig, 

Thank you for giving me the opportunity to submit a revised draft of my manuscript titled “The fracture resistance of 3D-printed versus milled provisional crowns: An in vitro study" (PONE-D-22-19019R2) to the Journal PLOS ONE. We appreciate the time and effort you and the reviewers have dedicated to providing valuable feedback on the manuscript, responded appropriately as noted below, and highlighted the changes within the manuscript.

Journal Requirements:

1) ‘’Please review your reference list to ensure that it is complete and correct. If you have cited papers that have been retracted, please include the rationale for doing so in the manuscript text, or remove these references and replace them with relevant current references. Any changes to the reference list should be mentioned in the rebuttal letter that accompanies your revised manuscript. If you need to cite a retracted article, indicate the article’s retracted status in the References list and also include a citation and full reference for the retraction notice’’.

Response: We would like to apologize for any inconvenient it might occur due to incorrect references list. Respectfully find in the manuscript with track changes the modified version of citations following the PLOS reference style outlined by the International Committee of Medical Journal Editors (ICMJE). The reference number 17 was eliminated. Also, reference number 7 was fully corrected. All other references were modified following “Vancouver” style. Thank you.

---

## [Editor Report · Decision Letter 3]

2 May 2023

The fracture resistance of 3D-printed versus milled provisional crowns: An in vitro study

PONE-D-22-19019R3

Dear Dr. Othman,

We’re pleased to inform you that your manuscript has been judged scientifically suitable for publication and will be formally accepted for publication once it meets all outstanding technical requirements.

Kind regards,

Mirza Rustum Baig

Academic Editor

PLOS ONE

Additional Editor Comments (optional):

Thank you for your valuable submission. Some amendments were suggested in the 'Discussion' section in the last decision letter sent (by the Editor). Please make sure you address these comments before the final editing of the article is completed.
---

## [Editor Report · Acceptance letter]

24 Aug 2023

PONE-D-22-19019R3 

The fracture resistance of 3D-printed versus milled provisional crowns: An in vitro study 

Dear Dr. Othman:

I'm pleased to inform you that your manuscript has been deemed suitable for publication in PLOS ONE. Congratulations! Your manuscript is now with our production department. 

Kind regards, 

on behalf of

Dr. Mirza Rustum Baig 

Academic Editor

PLOS ONE